Cattle and climate in Africa: How climate variability has influenced national cattle holdings from 1961–2008

Lunde Torleif Markussen 1 2 torleif.lunde@cih.uib.no
Lindtjørn Bernt 2
1 Bjerknes Centre for Climate Research, University of Bergen/Uni Research , Norway
2 Centre for International Health, University of Bergen , Norway
Xu Jianhua
Electronic publication date: 2013 Mar 19
Publication date: 2013
Volume: 1
Electronic Location ID: e55
Received 2013 Feb 9; Accepted 2013 Mar 3
Copyright: © 2013 Lunde and Lindtjørn
Copyright year: 2013
Copyright holder: Lunde and Lindtjørn
License: This is an open access article distributed under the terms of the Creative Commons Attribution License, which permits unrestricted use, distribution, and reproduction in any medium, provided the original author and source are credited.
License URL: https://creativecommons.org/licenses/by/3.0/

Keywords: Cattle, Climate, Africa, Malaria, Precipitation, Temperature

Funding: The Norwegian Programme for Development, Research and Education (NUFU) University of Bergen This work was made possible by the grants from The Norwegian Programme for Development, Research and Education (NUFU) and the University of Bergen. The funders had no role in study design, data collection and analysis, decision to publish, or preparation of the manuscript.

==============================
The role of cattle in developing countries is as a source of high-quality food, as draft animals, and as a source of manure and fuel. Cattle represent important contribution to household incomes, and in drought prone areas they can act as an insurance against weather risk. So far, no studies have addressed how historical variations in temperature and rainfall have influenced cattle populations in Africa. The focus of this study is to assess the historical impact of climate variability on national cattle holdings. We reconstruct the cattle density and distribution for two time periods; 1955–1960 and 2000–2005. Based on estimates from FAO and official numbers, we generated a time series of cattle densities from 1961–2008, and compared these data with precipitation and temperature anomalies for the same period. We show that from 1961–2008 rainfall and temperature have been modulating, and occasionally controlling, the number of cattle in Africa.

Introduction

Background

Since the 1960s there has been a period of climatic change with global land-surface temperature rising about 0.5–0.6 °C (Hansen, Sato & Ruedy, 2012). Although some studies have addressed effect of climate variability on cattle populations (Angassa & Oba, 2007; Cossins & Upton, 1988; Desta & Coppock, 2002; Oba, 2001), no studies have described how historical variations in temperature and rainfall influence cattle populations in Africa.

FAO estimates the number of cattle in Africa during the period 2001 to 2010 is twice the estimates for the years 1961–1970. But, how variations in the climate influence cattle depend on the ecological setting, and how variations in cattle influence the population, depend on the availability of alternative energy sources as well as the cultural setting. The role of cattle in developing countries is as a source of high-quality food, as draft animals, and as a source of manure and fuel (Scoones, 1992; Taddesse et al., 2003). Cattle represent important contribution to household incomes (Seo & Mendelsohn, 2006), and in drought prone areas they can act as an insurance against weather risk (Fafchamps & Gavian, 1997).

Most of the cattle in Africa are in arid and semiarid areas. In the forested humid areas of humid West-Africa, as well as Democratic Republic of Congo, the tsetse tolerant N’Dama and West African Shorthorn breeds are common. The most common cattle breed is, however, the East and West African Zebu, which make up the majority of African cattle (Deshler, 1963). Close to Lake Chad, the heat tolerant Kuri breed can be found, although the density has declined since the 1950s (Tawah, Rege & Aboagye, 1997), and in East Africa, the Sanga can be found on the western branch of the Great Rift Valley. In South Africa the Afrikander is common. The different cattle types probably represent mixtures of breeds introduced at various times (Deshler, 1963).

Many production systems supply water from ponds and rivers during the wet season, and the need for watering increases with higher temperatures (Seif, Johnson & Lippincott, 1979). The IPCC 2007 report concluded that changes in range-fed livestock numbers in any African region will be directly proportional to changes in annual precipitation (Intergovernmental Panel on Climate Change, 2007).

Rötter and van de Geijn (1999) discussed how changes in climate potentially can influence livestock;

• Feed production (Direct effect of CO2, temperature and precipitation)

• Animal health (Direct effect of feed production, heating through temperature, and watering via precipitation and evaporation)

• Diseases (Indirect effect of stress, parasites, and vector borne diseases).

The coming century, it is virtually certain temperatures will increase, and that the intensity of precipitation will change (Min et al., 2011). How the cattle has been, and will be influenced directly through climate variability, and indirectly through parasites and vector borne diseases is still uncertain. The lack of certainty in projected absolute changes in precipitation amounts and how cattle respond to climate, makes it difficult to to predict impacts of climate change. It is therefore necessary to understand the historical impact of climate on cattle before projecting future impacts and developing adaptation strategies (Hoffmann, 2010).

Feed production and consumption

The growth of plants are directly influenced by the atmospheric CO2 concentrations, with two metabolic pathways; C3 and C4 (Stokes et al., 2010). While the productivity is expected to increase for C3 plants, quality, productivity and digestibility is expected to decrease with increasing CO2 concentrations. C4 plants are probably less affected (Stokes et al., 2010). In the subtropical Australia it has been hypothesized that lower precipitation can be compensated for by the benefits of increased CO2 (Henry et al., 2012). The compensating effect in tropical Africa is uncertain.

A study by Seif et al. (Seif, Johnson & Lippincott, 1979) showed Zebu water consumption increased by 58% when temperature increased from 10 to 31 °C. This is only 2.8% increase per degree, if we consider this as a linear process. At higher temperatures, feed consumption decreases (Seif, Johnson & Lippincott, 1979) and fertility increases (Jöchle, 1972). Lower food consumption can be of importance in the dry season. This is in line with the perception of farmers in the savanna zone of central Senegal, who say low temperatures may lead to fodder shortage (Mertz et al., 2009).

Indirect consequences of climate

Although climate influence the vegetation in these semiarid environments, the coupling between climate and animal numbers might not be as straight forward as grass production and the need for watering; in tsetse infested areas, high temperatures might reduce the vector populations and cause a reduction in animal trypanosomiasis (Hall et al., 1984; Terblanche et al., 2008), in Nigeria increased rainfall has been linked to outbreaks of blackquarter (Bagadi, 1978), and in the eastern part of Africa, where east coast fever prevail, climate variability can be related to the survival and reproductive success of the tick Rhipicephalus appendiculatus (Branagan, 1973), and as the development of the Theileria parasite (Young & Leitch, 1981). Livestock also play a role in malaria transmission by creating favourable environments and blood meals for Anopheles arabiensis. We have previously shown that understanding fluctuations in cattle populations is important to assess the historical and future distribution of two of the most efficient vectors of malaria in Africa (Lunde et al., 2013a; Lunde et al., 2013b). Tirados et al. (2011) showed that a cattle herd of 20 heads outside a house reduced the number of Anopheles arabiensis landing on humans by 50%. It has also been speculated that certain malaria epidemics in India and Somalia can be explained by herds of livestock being decimated during drought years (Choumara, 1961; Cragg, 1923).

To quantify the historical impact of climate variability on cattle in Africa, we construct a statistical model which include precipitation (P), temperature during the rainy season (Tw), and temperature during the dry season (Td). We also adjust for armed conflicts and include a sigmoid-shaped Gompertz curve which represents an increase in infrastructure over time (number of herders/farmers, provision of wells/water stations, veterinary services) that allows an increased carrying capacity over time without population density-dependence.

Methods

The main aim of this paper is to quantify the effect climate variability has had on national cattle holdings from 1961–2008. To do so we specify a linear model under the assumption of normally distributed errors and constant variance: (1) Wny,c=β+m1⋅Ge(a,b)+m2⋅CFy,c+m3⋅Tdy,c+m4⋅Twy,c+m5⋅Py,c+ϵ

where β=Intercept

Ge(a,b)=Gompertz function with parameters a and b

CFy,c=Armed conflict weighted by cattle density within a country

Tdy,c=Five year weighted mean temperature anomalies in the dry season,spatially weighted by cattle density within a country

Twy,c=Five year weighted mean temperature anomalies in the wet season,spatially weighted by cattle density within a country

Py,c=Five year weighted annual mean monthly precipitation anomalies,spatially weighted by cattle density within a country

ϵ=Error.

In the following sections we explain how the spatial weights are constructed, the data sources, and corrections done to the data.

Construction of spatial weights

In 1963 Walter Deshler published a map of cattle distribution in Africa. The map is complete, except from two countries with large cattle populations; Ethiopa and Upper Volta (Burkina Faso). Data was also missing from Gabon and Spanish Sahara (Western Sahara), but these territories were probably empty of cattle. For Ethiopia and Upper Volta (Burkina Faso) we used FAO’s estimate of 2005 cattle density and adjust the totals to Faostat’s estimate for 1961. This process is described later.

We geo-referenced the raster map published by Deshler to a Miller Oblated Stereographic projection. Thereafter, the country borders, coastlines and rivers were manually removed, only leaving the dots in the maps. One point in the original map represents 5000 cattle (heads). In the rasterized version of the map, one point would consist of a group of pixels. The geo-referenced raster is a one band grayscale raster with values from 0 (black) to 255 (white). First, pixels with values grater than 200 were removed. Such a high threshold was chosen based on manually checking the distribution of representative dots. The remaining points could now be treated as probable candidates of being an observation of 5000 cattle. To automatically identify groups of points, we applied the Partitioning Around Medoids (PAM) algorithm (Kaufman and Rousseeuw, 1990). Since we knew the approximate total number of cattle in each country, and we also knew each point represent 5000 heads, the expected number of clusters was ≈ F A Otot,country⋅5000−1, where F A Otot,country is the FAO estimate of national cattle holdings. To speed up the algorithm we split the computation for each country in hexagonal tiles. After running the PAM-algorithm for all countries, except Ethiopia, Upper Volta, Gabon, and Spanish Sahara, we manually removed or added points which either were duplicates, or were not detected by the algorithm.

After the raster map had been converted to clean points, we used a spherical nonparametric estimator method to calculate point densities. Such kernel estimators were developed to omit problems with discontinuities of the estimates dependent on the bin positions. In this work we used a spherical kernel developed by Kevin Hodges (Hodges, 1996) (with power m = 1). This is a computational efficient kernel designed to derive storm track statistics. It is defined locally so that the influence of a point is restricted to a local region.

To choose a global smoothing parameter we maximize the cross-validation function suggested by Diggle and Fisher (Diggle & Fisher, 1985): (2) Γd(Cn)=−1n∑i=1nloge[fˆ−i(Xi,Cn)]

where (3) fˆ−i(Xi,Cn)=1n−1∑j≠inK(Xj⋅Xi,Cn).

Still the greatest value of the local, and hence global, smoothing parameter which is described later, is restricted by the grid spacing. If the spherical cap is too small, some points will not be included in the density estimation, and κ must therefore be restricted.

For the maps produced in this paper the value of κ=21907.45(κ˜=1.000046) which is equivalent to an arc bandwidth radius of 0.55∘. This parameter is then adaptively modified based on the ideas of a pilot density estimate and cross validation as described by Hodges (Hodges, 1996). If the smoothing parameter is (4) κ(κ˜=1+1/κ)

the local smoothing parameter is determined as: (5) κN,i=κNfˆp(Xi)gγ

where κN is the global smoothing parameter, fˆp(Xi) is the pilot estimate at each point Xi, g is the geometric mean of the pilot densities. The γ parameter is subjectively chosen to be 0.5 which Abramson (Abramson, 1982) showed (in the Cartesian domain) give lower bias than normal fixed bandwidth estimates.

After smoothing the cattle observation we normalize the densities to match 5000⋅n.

To estimate a comparable cattle density around year 2000 we converted the FAO observed bovine density (census data) (Robinson & Fao’s Animal Production and Health Division, 2011) to points, each point equal to 5000 animals. First, the FAO raster was converted to polygons using the Geospatial Data Abstraction Library (GDAL) (The Open Source Geospatial Foundation). In cases where the modulus of the sum inside the polygon is non-zero, the probability of sampling an additional point (Zhi+1) is the modulus divided by 5000. Next, we construct 50 realizations of the maps. Each time we sample ni completely spatial random points (Bivand, Pebesma & Gomez-Rubio, 2008; Pebesma & Bivand, 2005) within each polygon, and estimate the density as described earlier. Mostly, the observations from 2000 are aggregated to district level, and hence the observations do not have the same quality with respect to spatial distribution as those of Walter. The global smoothing parameter is held constant, while the local smoothing parameter will vary for each of the 50 estimates.

This method is used to provide a best-guess estimates of the cattle densities around 1960 and 2000 without making any assumptions about dependencies on land use or climate. There are two good reasons for doing this. First of all we do not know how the cattle distribution is related to climate within individual countries. Secondly; if we had already assumed that cattle distribution and density was dependent on climate or land use it would be hard to justify relating this data set to those variables.

Time series of national cattle holdings and spatially weighted time series of climate

FAOstat (FAO, 2011) reports the estimated number of cattle heads within a country from 1961. We relate this to the annual mean temperature and precipitation from University of Delaware air temperature and precipitation and repeat the same analysis with CRU v3.1. The data sets were interpolated to the same grid as the cattle densities using distance weighted interpolation. It should be noted that for example Madagascar, Somalia, and Ethiopia have very few weather observations. In countries with few observations, the results are less robust. Since the data from FAOstat is reported on national scale we need to aggregate the temperatures and precipitation to the same levels. To do this we use the newly constructed cattle densities. Each value inside the country (c) boundaries are given a weight (Wi,y,c) based on the cattle density. (6) Wi,y,c=Xi,y,c∑i=1nXi,y,c

where the cattle density in year (y) is linearly interpolated between 1960 and 2000.

The weighted mean temperature anomalies (T) or precipitation anomalies (P) for each country is then (given for T here): (7) Ty,c=∑i=1nTi,y,c⋅Wi,y,c.

Standardized anomalies can be calculated from the actual temperature or precipitation by dividing the difference from the mean on the standard deviation, or more specifically (x is actual temperature and n is the number of observations): (8) Ty,c=x−1n∑xx−1n∑x2n.

To account for the weather the past years we do an additional time smoothing with a kernel, K(9) K=[0.016,0.127,0.265,0.327,0.265].

And the new Ty,c becomes (10) Ty,c=∑i=04Ty−i,c⋅K[5−i].

Armed conflicts

To adjust for conflicts (C F) which might have influenced the cattle densities (Brück & Schindler, 2009), we use the armed conflict site data set from UCDP/PRIO. This data set contains year, coordinates (L) and radius in km (r) of conflicts (C F) from 1946 to 2005. On the same grid we define C Fi,j as a function of distance (D) from L and r. (11) CFi,j=D4r4

where C Fi,j > 0.

Allowing increased carrying capacity over time without population density-dependence

We introduce a sigmoid-shaped growth curve which represents an increase in infrastructure (number of herders/farmers, provision of wells/water stations, veterinary services). This function allows an increased carrying capacity over time without population density-dependence. We use a Gompertz function, and adjust the time and scale of the data. A description of the procedure is following in the next lines.

We normalize time (tn) from −2 to 2 (so that 1961 = −2 and 2008 = 2). This normalization is done based on the properties of the Gompertz function. The cattle numbers (W) from Faostat are also normalized (Wn) to range from min(Wn)=0 to max(Wn)=1: (12) Wn(t)=(max(W)−min(W))−1⋅W(t)+1−max(W)min(W)−1

where W(t) is the number of cattle at time t, and Wn(t) is the scaled number of cattle at time t.

Next, we estimate a and b using nonlinear weighted least-squares to optimize the function: (13) Ge(a,b)=a⋅e(b⋅e(−tn))

and (14) Wn=Ge(a,b)+ϵ.

Depending on the country, the cattle numbers reported by FAO might be based on estimates. Since these estimates are more unreliable than actual observations we want to give less weight to those. To define the weights we apply a two way search to find the minimum number of years since the last observation (Ω). For example if there were observations in 1999 and 2003, but not in 2000–2002, the weights for 1999, 2000, 2001, 2002 and 2003 would be 1−1, 2−1, 3−1, 2−1, 1−1.

Using Eq. (1) we use stepwise model selection by Bayesian information criterion (BIC) to estimate the model which explain most of the variance. A few cases suggested that war had a positive effect on cattle numbers. Since we believe this is unreasonable, war having a positive effect on national cattle holdings was not allowed in the model.

We assume errors follow a normal distribution, ϵ∼N(0,σ2), and test this assumption by applying a Shapiro-Wilk test of normality, as well as investigating the normal QQ plot of the residuals. To test for heteroscedasticity, we applied a Breusch-Pagan (Cook and Weisberg) test.

Data corrections

As mentioned, 1960 data was missing from Gabon, Spanish Sahara (Western Sahara), Ethiopia and Upper Volta (Burkina Faso). For Ethiopia and Upper Volta (Burkina Faso) we use FAO’s estimate of 2005 cattle density (Gridded livestock of the world (GLW) (Wint & Robinson, 2007)), and adjust the totals to Faostat’s estimate for 1961. For these countries. Since GLW was released, additional data has become available for Afder, Gode, Korahe, Warder, Fik, Degehabur, and Shiniele in Ethiopia (Central Statistical Authority, 2004). GLW is updated with this information. This data set should roughly give an estimate of the cattle distribution and density for 2000–2005. Since Ethiopia was classified as Ethiopia PDR in 1961 we used the total of Ethiopia and Eritrea in 2000 to match the 1961 Ethiopia PDR total. To make pseudo points for the four countries we randomly sampled (Bivand, Pebesma & Gomez-Rubio, 2008; Pebesma & Bivand, 2005) nearest integer of administrative zone totals divided by 5000 points in each zone.

For the present day estimates it should be noted that data for Mauritania was missing. FAO does report the estimated total, and to estimate the density for Mauritania we distribute the total in the areas which are not reported as zero. There are two major areas in Mauritania which are likely to have cattle. The major area is to the south, while a smaller area is located around 21.5 North–6.6 degrees East. In the latter area we assume the density to be approximately equal to the density on the Mali side of the border, while the remaining is equally distributed in the Southern area.

Non-technical summary of the methods

We estimated a continuous surface of cattle densities and distribution from point observations. From this data we calculated annual mean precipitation, and dry and wet season temperature anomalies where the cattle were present. These time series were correlated with the official cattle holdings for each country using a linear model, giving more weight to actual observations, and less weight to estimates. In addition, we have included the Gompertz function to account for adaptation and growth. We also adjust for armed conflicts which has been important for cattle numbers in, for example, Mozambique (Brück & Schindler, 2009).

Results

Figure 1 shows the estimated cattle distribution and density in the 1960s and 2000s, and their difference over the mean density. In general, most areas have more cattle today than in the early phase of global warming. But there are some exceptions. The declines are mainly observed in Northern Somalia, and Northern Kenya, around the Niger river, Mauritania, parts of South Africa, Mozambique, Namibia, and Madagascar. Although there are large variations, the main signal is that dry areas have seen a reduction in the amount of cattle opposite to wetter areas, which have seen an increase.

Figure 1 Cattle density and distribution 1960s and 2000s.

Estimated cattle (heads per square kilometre) density around 1960 (upper left), 2000 (upper right) and difference (2000–1960, bottom) relative to the mean.

Figure 2 shows the sign of the slope for precipitation (P), wet season temperature (Tw), and dry season temperature (Td). Only values where the degree of confidence is greater than 95%, and the model could explain more than 30% of the variance are shown. In most of the countries where precipitation is significantly correlated with cattle numbers, increased annual rainfall is associated with increased cattle numbers. The exceptions, which have notable cattle populations, are Cameroon, Nigeria, and Benin, countries which have warm and relatively humid climates.

Figure 2 Effect of climate variability on cattle holdings.

Upper panel (left to right): Sign of slope for precipitation (P), wet season temperature (Tw), and dry season temperature (Td). Positive values means increased precipitation/temperature is associated with increased number of cattle and vice versa. Middle panel (left to right): Percent variance explained by precipitation, wet season temperature and dry season temperature. Lower panel (left to right): Total variance explained by the model, variance explained by variability in Gompertz function, and variance explained by climate variability.

On the other hand, the influence of temperature during the wet season demonstrate a more diverse pattern. Temperature can influence cattle through direct heating, through vector borne diseases, and by modulating evaporation. However, the most dominant factors controlling evaporation is the vapor content in the air and the turbulence which can transport vapor away from the surface. There are eleven countries where temperature explain more than 10% of the variance, where warmer wet season temperatures have a positive impact in Lesotho (10%) and Ghana (17%), and a negative impact in Zimbabwe (7%), Uganda (5%), Benin (17%), Mali (13%), Senegal (11%), Mauritania (13%), Libya (17%), Morocco(/Western Sahara) (30%), Liberia (10%), Gambia (10%), and Tunisia (21%).

The pathway from temperature to cattle in these countries is probably diverse. For example in Tunisia and Morocco theileriosis is influenced by temperature. In the drier and warmer countries, the need for water increases as the temperature increases.

Outside the rainy season, temperature can interact differently. At higher temperatures, feed consumption decreases (Seif, Johnson & Lippincott, 1979) and fertility increases (Jöchle, 1972). Lower food consumption can be of importance in the dry season. According to the model, Western Sahel seems to be especially sensitive to the dry-season temperature. This is in line with the perception of farmers in the savanna zone of central Senegal, who say cold temperatures may lead to fodder shortage (Mertz et al., 2009).

In the least climate sensitive countries, the Gompertz model can explain most of the variance. We interpret the greatest climate sensitivity will be seen where resource limits are reached, and the resource limits are then modulated by climate (Fig. 3A). Thus, we expect the present day climate insensitive countries to be more vulnerable to climate variability as the cattle populations converge toward the carrying capacity limit; One way to adapt to climate variability, from season-to-season and year-to-year, is to move cattle to new areas. As the number of cattle increases, there will be more competition for food and water, and the strategy of moving cattle might be less successful if most areas are already occupied. In this case, use of concentrate feeds might be a viable alternative, while the access to water might still make the cattle populations vulnerable.

Figure 3 Climate sensitivity.

Left: Relationship between the variance explained by the Gompertz function and climate. In countries where the number of cattle fit well with the Gompertz curve there is little climate sensitivity. Where the carrying capacity is close to its limit the cattle populations will fluctuate around the mean and hence the Gompertz function will not explain the variance. In these countries climate variability is the major source of variation. Vertical lines are box-and-whisker diagrams plotted at quantiles 0–0.1 to 0.9–1 by 0.1. Black dot indicate median for each quantile. Right: The effect of one standard deviation increase in precipitation vs annual mean rainfall (1961–1990). Y-axis is in %, and show the change in cattle numbers relative to the mean number of cattle (1961–1990). A value of 50 on the y-axis would indicate one standard deviation of rainfall (over five years) would increase the national cattle holdings by 50% relative to the mean. A value of −50 would indicate a reduction of 50%. Vertical lines are box-and-whisker diagrams plotted at quantiles 0–0.25 to 0.75–1 by 0.25. Black dot indicate median for each quantile.

Although temperature is important, precipitation show a more consistent pattern, with a positive effect in the drier countries, and a negative effect in wetter countries (Fig. 3B). We obtained similar results as the ones shown in Fig. 3B when using a mixed effect model (Bates, Maechler & Bolker, 2011). In this analysis, countries were classified based on the UNEP index of aridity (Middleton, United Nations Environment Program & Thomas, 1997), taking into account the potential evapotranspiration and average annual precipitation. The mixed linear model, with the same form as the linear model described earlier, was repeated for cattle belonging to each of aridity classes with country as a random variable. In this analysis, the fixed effect of one standard deviation increase in precipitation were; Arid: 0.60 (CI 95% 0.47, 0.74), Semi-arid: 0.11 (0.01, 0.22), Dry subhumid: 0.14 (0.07, 0.22), Subhumid: −0.01 (−0.07, 0.05). No clear associations were found between temperature anomalies and national cattle holdings.

Discussion

The cattle populations which are most sensitive to climate variability are located in arid regions. In the IPCC 2007 report it was stated that “… changes in range-fed livestock numbers in any African region will be directly proportional to changes in annual precipitation.” This is only true in dry environments, and that in wetter environments, increased precipitation has no, or negative impacts on the national cattle stocks.

The observed negative association between precipitation and national cattle holdings in humid countries could be the effect of cross-border movement of livestock. We show that, while the Sahelian nations Niger and Chad show the expected positive relationship between rainfall and cattle numbers, their southern neighbours (Benin, Nigeria, Cameroon and CAR) have negative relationships. Figure 1 shows that the majority of cattle both these groups of countries are close to the borders between the former and latter groups. Since nomadic cattle herding or transhumance is the dominant form of cattle herding in this region, a likely explanation is that cattle are driven south during periods of drought in the Sahel to areas with higher rainfall. In Mali, cattle are less frequently herded across the southern borders but instead are driven from the Sahel into higher rainfall regions further south within the country. This may explain in part why Mali has no clear relationship with rainfall, as the main effect of drought may be a shift in cattle further south within the country. Unfortunately the data presented here do not allow quantifying cross-border movement. It is also possible that the negative response to precipitation in humid areas can be explained by precipitation driven variability in the tsetse-fly populations, with more efficient transmission of trypanosomes in wet years. To establish any associations between variability in the number of tsetse flies and cattle, there is a need for historical datasets describing how the number of tsetse flies has varied in space and time.

Due to lack of consistent weather records for parts of Africa, and uncertainty in the estimated cattle densities, the absolute estimated effect of preceding precipitation and temperature anomalies on national cattle holdings should be interpreted with care. The consistent response in dry and humid regions does however make physical sense, with long wet or dry periods being the main factor controlling water and food availability, and in the end, cattle. We have not addressed if the declines and increases in cattle numbers are reflecting planning, by for example slaughtering and selling more cattle in dry periods, or if the declines in cattle numbers in dry periods can be attributed to natural mortality due to lack of food and water. These are important aspects which should be further investigated and documented, since the response to long term (30 years) changes in the climate might be different from the responses to short term fluctuations (2–3 years).

Since this is a statistical model with its limitations, it is therefore optimistic to extrapolate these relationships into the future. Instead we will show the expected changes in precipitation in the next century, and combine these maps with information about where cattle was present in the 1960s. We look at three of the Representative Concentration Pathways (RCP), where RCP 2.6 (Image, 14 models) showing future climate with strong mitigation, RCP 4.5 (miniCam, 18 models) is an in-between scenario and RCP 8.5 (Message, 16 models) assumes no mitigation. Figure 4 shows the expected response in precipitation under the four different scenarios. To minimize the effect of multidecadal variability we have used three averaging periods, baseline (1961–2000), near future (2006–2050), and distant future (2051–2100). Under the scenario where no mitigation takes place (Message), the Southern part of Africa will have a reduction in mean annual precipitation of 0.1–0.2 standard deviations in 2051–2100. Under the Image scenario the signal is weakened and the agreement between the models is lower. The miniCam scenario lies in between. The Southern part of Africa is one of the areas with a relatively high sensitivity to precipitation, and it is very likely that the cattle populations in this region will be negatively affected if no mitigation takes place. However, the increased CO2 can reduce the impact of decreased rainfall, by increasing the soil nutrient availability. It is also likely that mean precipitation will decrease in parts of Mauritania, Mali and Senegal. The opposite is true for Eastern Africa and Eastern Sahel. These areas have shown little sensitivity to precipitation the last 50 years, and it is likely to very likely that the annual precipitation will increase with no mitigation. The signal is weaker under the Image and miniCam scenarios.

Figure 4 Future precipitation patterns.

The mean expected changes from 1961–2000 in precipitation under three different climate scenarios. Shading indicate standard deviations, black contours show the presence of cattle in the 1960s. Only values where more than 66% of the models agree are shown. Black dots indicate more than 90% of the models agree on the sign of change.

It is interesting to note that the East African countries are less sensitive to climate variability. If new areas are utilized for cattle production over time, like the Ethiopian case (Funk et al., 2012), the resulting increased carrying capacity of a country will wash out the effects of climate variability in the data. In the analysis presented here we suggest the national cattle holdings in Ethiopia have not been influenced by climate variability. However, the cattle populations in the Borana pastoral production system in Ethiopia have been strongly influenced by rainfall variability and trends (Angassa & Oba, 2007). It is therefore important to remember our analysis is restricted to nations.

The concept of carrying capacity is briefly addressed in the 2007 IPPC report, part III (Intergovernmental Panel on Climate Change, 2007). Carrying capacity is a key to understand how cattle will be influenced by changes in the climate. Adaptation on national scales can happen through utilization of new areas, but at some point there will be few new areas to use, and the vulnerability to climate will most likely increase.

Additional Information and Declarations

Competing Interests

Author Contributions

Data Deposition

The authors declare that they have no competing interests

Torleif Markussen Lunde conceived and designed the experiments, performed the experiments, analyzed the data, contributed reagents/materials/analysis tools, wrote the paper.

Bernt Lindtjørn conceived and designed the experiments, wrote the paper.

The following information was supplied regarding the deposition of related data:

ftp://ftp.uib.no/pub/gfi/tlu004/OMaWa/

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
