# Peer review of "Cattle and climate in Africa: How climate variability has influenced national cattle holdings from 1961–2008"

_PeerJ, doi:10.7717/peerj.55_

## Round 0.1 · original submission · Major Revisions

I now have two reviews of your manuscript from experts. One of the experts recommends your manuscript for publication after minor revision, and another recommends your manuscript for publication after major revision. After careful consideration based on the reviewers’ comments, I has decided that your manuscript requires a careful revision before it can be accepted for publication.

Reviewer 1 ·

Basic reporting

Analysis of impact of future climate change on populations including that of herbivores such as cattle would certainly benefit from retrospectively understanding responses to past climate variables such as rainfall and temperature variability. More importantly, the scaling of data series varied from local to regional on effects of climate change is useful for improving the predictions. The authors of the above manuscript have made some effort in developing a mathematical model to capture varieties of variables that would influence the biological productivity of the grazing lands and subsequently, cattle population variability across geographical and political regions in Africa. Using time-series data the authors dialogued the model to understand its’ power of predictions based on their mathematical assumptions in model constructions as well as related ecological responses. Some efforts have also been made to understand if the behaviour of herd movements would be captured by the model. In that sense the authors have made an important contribution to better understanding of impact of future and past climates on variations of regional cattle populations. However, the novelty of their contribution and the functionality of the model can only be judged from their assumptions both in model construction and associated independent variables.

Experimental design

While this referee found the importance of this manuscript to increasing scientific understanding of the regional impact of the climate on populations of cattle, there remain important clarifications to be made. These will be itemized.
1. The authors beginning with the abstract appear to argue a direct relationship between the variability of climatic variables (such as temperature and moisture) and prevalence of mosquito and yet, this item has not been well developed in the paper.
2. If temperature and rainfall influence on cattle population is only occasional, then there has to be strong arguments which biological factors would assist in such deductions.
3. The opening statement on P. 2, is too strong and further down a contradiction is implied. This need to be rephrased.
4. Statements such as carrying capacity need to be applied carefully and often these are located in wrong places in the manuscript. Shift them to where the statements make a better sense. E.g. P. 3, the statement “The role of cattle…” need to be moved to the background section. Further on P. 10, the last para. The statements on carrying capacity appear to be in the wrong place. It could be incorporated into the methods section as assumptions. For example.
5. In the model, why is raiding a necessity? At the regional scale, raids, would not influence changes in cattle population, considering that it involves moving the population locally from one community to another. If however, regional scale wars such as that between Ethiopia and Eritrea have been assumed it would be interesting to see if between different periods of conflicts cattle populations in the border regions of Sudan or Somalia had changed. This variable as it stands is unlikely to be useful in the model.
6. As an alternative raiding factor in the model, the authors should consider prevalence of disease vectors such as tsetse flies. It is a well established fact that cattle distribution in Africa is influenced by tsetse, marked by the belt of the buffer zones between the sub-humid and semi-arid zones. This might explain the negative responses in their predictions.

Validity of the findings

7. The authors could have made fundamental contributions if they took into consideration the ecological variability of the continent more seriously. For example, what about making generalized sampling protocols by: dividing the continent into arid, semi-arid and sub-humid and run different sub models. The aim then is to clarify the assumptions as to which factors were driving the cattle population. Currently, we have statements that suggest that some regions were sensitive and others insensitive to rainfall variability. One would need better explanation than those offered. See the 3 para on P. 3.
8. On P. 9, 2nd para. The statement “Thus, we expect the present climate…” should explain why the countries presently insensitive to climate variability are expected to be affected more in the future.
9. In your discussions, present explanation and interpretation as opposed to repeating results. For example, the cross border movements of herds have important implications. It is an habit that pastoralists in the regions have developed even prior to colonial borders. Yet, these shifting would show changes between seasons, while greater shifting into the sub-humid Sudanian savannas might be indicative of long-term desiccation.
10. The section on P. 10, from second, para. need to be rewritten with the above comments in mind.
11. Finally, the reference [6] given as an end note on P. 10, does not seem to be consistent with the others.

Additional comments

No Comments

Reviewer 2 ·

Basic reporting

There are minor problems in the figure captions which better be corrected.

Experimental design

No Comments

Validity of the findings

Minor problems:
1. the author should put more efforts to make the work more accessible to general research comminity the correlationship interpretation between the indirect factors (precipatation and temperature) and the modulated and some times controlled cattle numbers/distribution;

Specifically, the manuscript introduces a number of variables, e.g. in equation (1), but notation is not always well defined (e.g. is the symbol b of "intercept" exactly the same as the parameter b in Gompertz function?), dimensions/units are not given for any of the introduced quantities; dimensions/units for non-trivial quantities should be given when they first appear. Alternatively, a list of symbols with units would be helpful. The unit ambiguity and associated physical interpretation problems can be exemplified.

2. Since missing or not so much reliable data have been employed, both for the cattle numbers and for the temperature/precipatation data for some countries in some special time intervals, different interpolation /extrapolation or missing data handling strategies may have significant artificial effects on the conclusion. The author should validate the robustness of the methods used especially when there are missing or artificial data;

3. By saying "modulating"(say, conclusion in abstract), espcially when both temperature and precipitation are indirect effectors and therefore the "effect mechanism" may be complicated (which does not have to be interpreted), the current analysis failed to interpret the possible "time lag" between the possible causes and the effect(s). (This is only a piece of optional suggestion, without interpreting this, the current analysis is already sound enough)

Additional comments

The manuscript is generally well written and clearly presented. (accept, minor revision)

---

## Round 0.2 · accepted · Accept

I have enjoyed working with you on this paper and look forward to reading your other papers in the future.